# Ovarian Clear Cell Carcinoma and Mature Cystic Teratoma Transformed to PNET and Carcinosarcoma: A Case Report with an Immunohistochemical Investigation

**DOI:** 10.3390/biomedicines10030547

**Published:** 2022-02-24

**Authors:** Mikiko Matsuo, Chiemi Saigo, Tamotsu Takeuchi, Akane Onogi, Naoki Watanabe, Shinsuke Aikyo, Hiroshi Toyoki, Hiroyuki Yanai, Takuji Tanaka

**Affiliations:** 1Department of Tumor Pathology, Gifu University Graduate School of Medicine, Gifu 501-1193, Japan; y2111023@edu.gifu-u.ac.jp; 2Department of Pathology and Translational Research, Gifu University Graduate School of Medicine, Yanagido, Gifu 501-1193, Japan; chiemi3150@yahoo.co.jp (C.S.); takeutit08@gmail.com (T.T.); 3Research Center of Diagnostic Pathology, Department of Diagnostic Pathology Gifu Municipal Hospital, Gifu 500-8513, Japan; aonogi0108@yahoo.co.jp (A.O.); naoki@watanabe.name (N.W.); 4Department of Obstetrics & Gynecology, Gifu Municipal Hospital, Gifu 500-8513, Japan; oretti_shinsuke@yahoo.co.jp (S.A.); toyohiro370601@yahoo.co.jp (H.T.); 5Department of Pathology, Okayama University Hospital, Okayama 700-8558, Japan; yanaih@md.okayama-u.ac.jp; 6Center of Genomic Medicine, Department of Diagnostic Pathology, Gifu Municipal Hospital, Gifu 500-8513, Japan

**Keywords:** ovary, mature cystic teratoma, PNET, carcinosarcoma, clear cell carcinoma, germ cell tumor, somatic-type malignancy, transformation, immunohistochemistry

## Abstract

Ovarian tumors include neoplasms derived from somatic cells and germ cells, including teratoma. Sometimes, tumors of the somatic cell type may develop from teratoma, causing diagnostic perturbation. We experienced a case of a tumor composed of several types of tissue in the ovary with a teratoma. When findings of teratoma and somatic tumor coexist in an ovary, it is difficult to differentiate whether a somatic tumor was mixed with a teratoma or a teratoma unitarily caused transformation to a somatic cell tumor. A 72-year-old Japanese woman (gravida, 3; para, 1) presented to our hospital with severe constipation and frequent urination, and a large intrapelvic tumor was detected by computed tomography (CT). Soon after admission, ultrasonography (US) and magnetic resonance imaging (MRI) revealed a large multilocular cystic tumor on her left ovary. Based on the clinical diagnosis of ovarian cancer, she underwent a left ovariectomy, appendectomy, and partial omentectomy. We observed an ovarian tumor consisting of teratoma, primitive neuroectodermal tumor (PNET), adenocarcinoma, various types of sarcomas, and clear cell carcinoma on the H and E-stained sections. The component of clear cell carcinoma showed a nuclear positive reaction against PAX8 and napsin A, as well as a loss of ARID1A, suggesting typical endometriosis-derived clear cell carcinoma. On the other hand, the expression of ARID1A was maintained in teratoma, PNET, non-specific adenocarcinoma, and various types of sarcomas, suggesting that these tumors had an origin different from that of clear cell carcinoma. These findings indicated that the ovarian tumor of this patient contained a clear cell carcinoma derived from a somatic cell and a teratoma that transformed to a wide variety of somatic cell types of tumors, which coexisted on one ovary. The appropriate use of immunohistochemistry was diagnostically effective in this case.

## 1. Introduction

According to the WHO classification [1], ovarian neoplasms have a variety of histological types and include those from epithelial, mesenchymal, sex cord-stromal, germ cell, and mixed epithelial and mesenchymal origins. They also possess benign and malignant phenotypes. Regarding mixed cell types, ovarian carcinosarcoma (OCS) rarely develops in elderly women, representing between 0.5% [2] and 6% [3] of all primary ovarian malignancies. Histopathologically, OCS contains carcinomatous and sarcomatous neoplastic cells [1,4]. The former is usually serous, endometrioid, mucinous or clear cell carcinoma. The latter originates from ovarian mesenchymal tissue (homologous) or extraovarian mesenchymal tissue (heterologous) [1]. Most patients with OCS are diagnosed in advanced stages, and the prognosis is generally poor. In addition, there is a unique tumor type, called “teratoid carcinosarcoma” [5,6] or “teratocarcinosarcoma” [7]. Additionally, it is known that mature teratoma can transform to carcinoma [1], a primitive neuroectodermal tumor (PNET) [8], sarcoma [1], or carcinosarcoma [9]. Ovarian tumors can be derived from somatic or germ cells [10,11,12]. Occasionally, teratomas can develop somatic tumors, which can make diagnosis difficult.

We recently experienced a multicomponent tumor that contained a somatic type of clear cell carcinoma, mature teratoma, and a variety of tumors transformed from teratoma components in a single ovary. Immunohistochemistry was effective for diagnosing each component in the tumor. The clear cell carcinoma component was PAX8-positive and napsin A-positive, and ADID1A loss was observed, indicating typical somatic cell-derived clear cell carcinoma. On the other hand, teratoma and teratoma-derived tumors, including PNET, non-specific adenocarcinomas, and various sarcomas, retained ARID1A expression. We herein report the histopathological and immunohistochemical findings of the patient’s tumor and discuss its pathogenesis/histogenesis.

## 2. Case Report

A 72-year-old Japanese woman (gravida, 3; para, 1) presented to our hospital with severe constipation and frequent urination due to a large intrapelvic tumor found by her family doctor with computed tomography (CT) (Figure 1a). Her past history included the removal of a right ovarian cyst (details unknown) and the treatment of Graves’ disease (details unknown) at 40 years of age. Menopause had occurred 19 years previously. Her family medical history was unremarkable. A physical examination yielded no significant findings. Serum tumor markers, including CA125 (18.9 U/mL), CA19-9 (7.2 U/mL), AFP (3.4 ng/mL), CEA (4.5 ng/mL), CA15-3 (7.2 U/mL), and SCC (1.7 ng/mL), were also within normal limits. Ultrasonography (US) showed an enlarged left ovary measuring 105 × 138 × 128 mm with multicystic lesions, where the septum of the cysts was thick, and a solid part was seen in part of the wall (data not shown). Magnetic resonance imaging (MRI) revealed a multilocular cystic tumor in the left ovary (Figure 1b,c). The inside of the tumor showed a non-uniform signal, and on the left outer side of the cyst wall, a solid part showed an intermediate signal on T2-weighted imaging and an abnormal signal on diffusion-weighted imaging (Figure 1b,c). These findings suggested a left ovarian cancer, and a left ovariectomy was performed, along with an appendectomy and removal of the greater omentum, although we did not observe peritoneal metastasis. During the operation, we noticed the left ovary with a cystic tumorous lesion was attached to structures. The uterus and right ovary showed no remarkable changes. During the operation, part of the capsule of the ovarian cystic tumor ruptured; however, imaging examinations confirmed that there was no metastasis or dissemination.

Grossly, the left ovary was multilocular cystic (Figure 2a). Part of the wall was fragile and collapsed inside (Figure 2b). Some parts contained teratomatous elements, including keratin, waxy sebaceous, gelatinous material and fatty nodularity (Figure 2b). Histopathologically, the fragile and collapsed part showed compact nests and sheets of atypical glandular cells with clear cytoplasm and hobnail nuclei, suggesting clear cell carcinoma (Figure 3a), and immunohistochemically nuclear positivity of PAX8 (Figure 3b) and napsin A (Figure 3c) was observed. The clear cell carcinoma was negative for ARID1A (Figure 3d). In other parts of the ovary, skin tissue (Figure 4a), including squamous epithelium, sebaceous glands, hair follicles, bone, and fat, was found, and this lesion was diagnosed as mature cystic teratoma. This ovarian cystic tumor also included a small round-cell tumor (Figure 4b). The small round-cell tumor was positive for NKX2.2 (Figure 4c), CD56 (Figure 4d), and glial fibrillary acidic protein (GFAP, Figure 4e), suggesting PNET [13]. A few round-cell tumor cells were weakly positive for CD99 (Figure 4f), but neural tube-like structures were not observed. In the other part of the cystic tumor, there was a sarcomatous component where atypical spindle-shaped cells densely proliferated (Figure 4g) and that partly showed cartilage differentiation (Figure 4g), with immunohistochemistry revealing S100 positivity, suggesting chondrosarcoma. Furthermore, an adenocarcinoma component (Figure 4h) was scattered in the sarcoma component. These adenocarcinoma cells were positive for the expression of AE1/AE3 but negative for PAX8 (Figure 4i) and napsin A. They were positive for ARID1A (Figure 4j). Nuclei of PNET and sarcoma cells were positive for ARID1A. 

Based on these findings, the ovarian tumor was diagnosed as a coexistence of ovarian mature teratoma showing transformed malignancies, PNET, and carcinosarcoma, as well as somatic-type malignancy and clear cell carcinoma. The tumor was classified as stage IC1 (pT1c1pNXpM0) according to the International Federation of Gynecology and Obstetrics (FIGO) 2014 classification. After the operation, six courses of TC (paclitaxel + carboplatin) chemotherapy were planned. After the end of three courses, TC chemotherapy was stopped due to severe side effects. However, the patient experienced an uneventful clinical course without recurrence or metastasis. Regular follow-up was continued in our hospital without signs of relapse in the 12 months since diagnosis.

## 3. Discussion

An extremely rare case of coexistence of somatic- and teratoma-derived tumors in the ovary of a 72-year-old female was presented. Ovarian mature cystic teratoma of germ cell deviation consists of tissues derived from all three germ-cell layers. PNET arising from a germ cell tumor can develop from malignant transformation of mature teratoma along ectodermal lines [8,14]. Transformation of a cystic benign teratoma of the ovary into a “carcinosarcoma” has very rarely been reported [9,15]. Additionally, clear cell carcinoma was noted in the same ovary containing a mature cystic teratoma. In this case, the evidence of the germ-cell origin of PNET and carcinosarcoma and somatic malignancy of clear cell carcinoma is supported by immunohistochemistry with an ARID1A antibody. Secondary malignant transformation within a mature teratoma is a much rarer occurrence, estimated as less than 2% of all such lesions [16]. Adenocarcinomas are the second most common malignancies arising within mature teratomas. Sarcomas alone or in combination with squamous cell carcinoma have been described as arising in a mature cystic teratoma. To the best of our knowledge, no case of sarcoma arising in association with adenocarcinoma has been described before.

To understand the pathogenesis of this ovarian tumor, we determined the cellular origin(s) of several of its components. We considered that the tumor was either (i) mature teratoma transformed to a somatic tumor; (ii) teratosarcoma; or (iii) a coexistence of clear cell carcinoma and mature teratoma transformed to a somatic tumor. Immunohistochemically, clear cell carcinoma was negative for ARID1A, suggesting ARID1A mutation; however, other tumor components were positive for ARID1A, suggesting wild-type ARID1A. These findings suggest that clear cell carcinoma and other components came from a different clone. Furthermore, clear cell carcinoma was immunohistochemically PAX8-positive, suggesting Müllerian origin [17,18], and the ARID1A mutation is often observed for endometriosis-related adenocarcinoma [12,19]. Such evidence also supports our conclusion that the clear cell carcinoma component of this ovarian tumor was not of teratomatous origin but of somatic cell origin.

Additional findings in this case include the immunohistochemical expression of ARID1A in clear cell carcinoma (negative stainability in nuclei), PNET (positive in nuclei), and carcinosarcoma (positive in nuclei), as well as ARID1A located on chromosome 1p36.11, which is a core member of the SWI/SNF complex [20]. This complex plays a crucial role in oncogenesis, and thus, ARID1A is considered a tumor suppressor gene. The mutation of this gene is suggested to be involved in the onset and progression of several types of malignancies. Next-generation sequencers have enabled genome-wide analyses, and mutation of ARID1A in ovarian clear cell carcinoma and ovarian endometrioid carcinoma was first reported in 2010 [12,21]. Since then, ARID1A mutations have been found in many human cancers and sarcomas [20,22]. At present, no metastasis/recurrence has been noted in our case 12 months after surgery followed by chemotherapy. The ARID1A expression of the tumors, which is strongly associated mismatch repair deficiency, may be involved in the prognosis of this tumor [23,24]. Therefore, we planned a regular and lifelong follow-up.

## 4. Conclusions

Our case is extremely rare because ARID1A-negative somatic cell-derived clear cell carcinoma coexisted with mature cystic teratoma and various tumors (PNET-positive for NKX2.2; CD56; glial fibrillary acidic protein (GFAP); non-specific adenocarcinomas with AE1/AE3 positivity; and various types of sarcomas, including chondrosarcoma with S100 positivity), which were transformed from teratoma in a single ovary with nuclear-positive ARID1A. Clear cell carcinoma was immunohistochemically negative for ARID1A, suggesting ARID1A mutation, which is common in endometriosis-related adenocarcinoma, and PAX8-positive staining being suggested its Mullerian origin. Taken together, this case is an exception and a complicated form of ovarian tumor derived from different mechanisms. This case also highlights the significance of appropriate immunohistochemical study to identify the cellular origin and histogenesis of ovarian cancer.

## Figures and Tables

**Figure 1 biomedicines-10-00547-f001:**
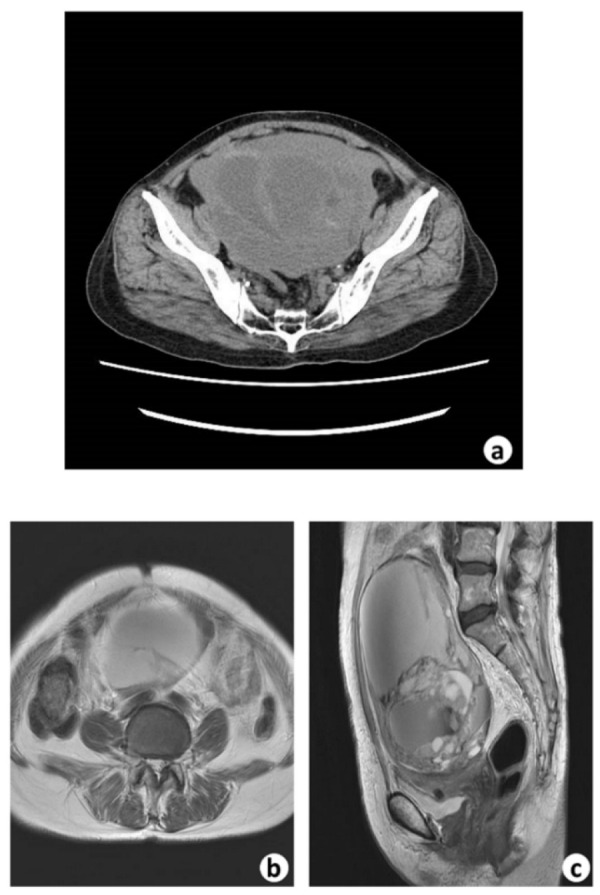
CT and MRI images. (**a**) Pelvic CT shows a large cystic tumor occupying the pelvic cavity. (**b**) T2-weighed axial MRI of the pelvis showing a large left ovarian cystic tumor. (**c**) T2-weighed sagittal MRI of the pelvis showing a large multilobular ovarian cystic tumor.

**Figure 2 biomedicines-10-00547-f002:**
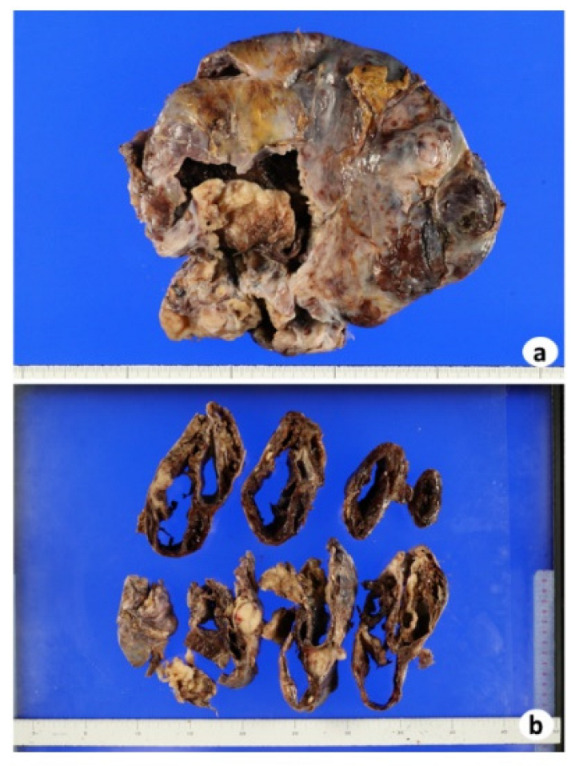
(**a**) Macroscopic view of a left-ovarian large cystic tumor that was surgically removed. (**b**) The cut surface of the tumor is multilocular with hemorrhage and necrosis.

**Figure 3 biomedicines-10-00547-f003:**
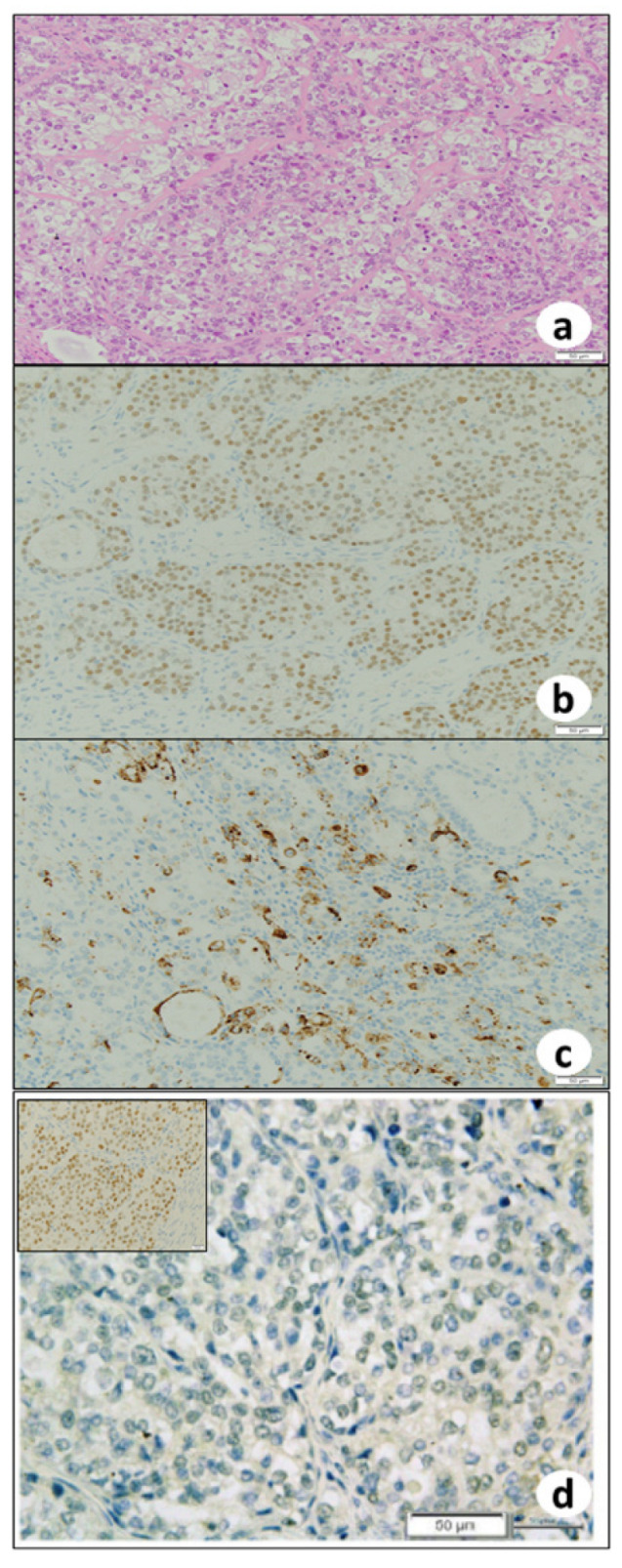
Histopathologically, carcinoma components include (**a**) clear cell carcinoma with compact nests and sheets of cells. Clear cell carcinoma showing diffuse nuclear positivity against (**b**) anti-PAX8 and (**c**) anti-napsin A antibodies. (**d**) There is no nuclear stainability of ARID1A in clear cell carcinoma. (**a**) Hematoxylin and eosin staining; (**b**) PAX8 immunohistochemistry; (**c**) napsin A immunohistochemistry; and (**d**) ARID1A immunohistochemistry. (**a**–**d**) bar = 50 μm. Insert in (**d**) is positive control of ARID1A.

**Figure 4 biomedicines-10-00547-f004:**
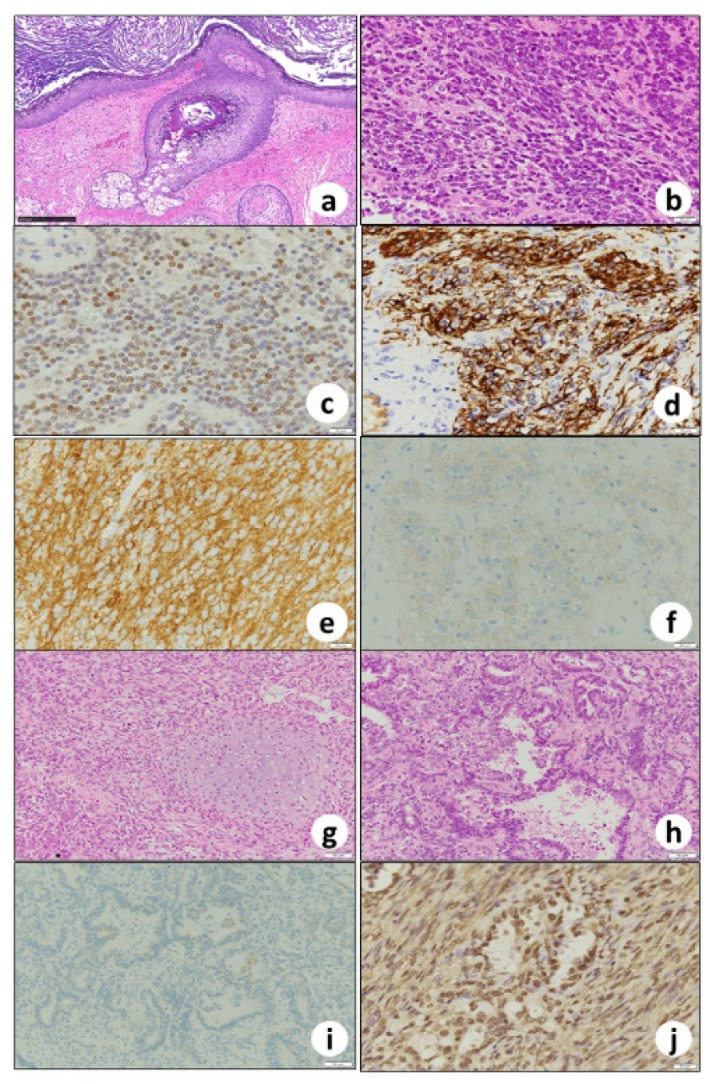
The cystic lesion is lined with (**a**) epidermis with skin appendages. Histopathology and immunohistochemistry of (**b**) small round-cell tumor nuclear-positive for (**c**) NKK2.2, membranous-positive for (**d**) CD56, cytoplasmic-positive for (**e**) GFAP, and weakly membranous-positive for (**f**) CD99, suggesting PNET. Histopathology of (**g**) spindle cell sarcoma component, partly differentiated to chondrosarcoma and (**h**) non-specific adenocarcinoma component. Adenocarcinoma component is negative for (**i**) anti-PAX8 but positive for (**j**) ARID1A. (**a**,**b**,**g**,**h**) Hematoxylin and eosin staining, immunohistochemistry of (**c**) NKK2.2, (**d**) CD56, (**e**) GFAP, (**f**) CD99, (**i**) PAX8, and (**j**) ARID1A. Bars: (**a**) 250 μm; (**b**–**f**) 20 μm; and (**g**–**j**) 50 μm.

## Data Availability

Not applicable.

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
