# Peer review of "Ovarian Clear Cell Carcinoma and Mature Cystic Teratoma Transformed to PNET and Carcinosarcoma: A Case Report with an Immunohistochemical Investigation"

_biomedicines, 2022, doi:10.3390/biomedicines10030547_

Round 1

Reviewer 1 Report

Thank you for the opportunity to review this case report on an unusual mixture of histopathologies within an ovarian specimens. The case is clearly presented. There is logical flow of the diagnositic work up by pathology and the ultimate conclusions.

I have the following clinical questions

  1. At the Operation there was no description of the uterus and other ovary. Were these structures present or not?
  2. Was the left ovary attached to structures or freely mobile
  3. In an older woman I would have thought there would be peritoneal biopsies or node sampling/dissection? Explain the decision not to do so.
  4. Sometimes Calcium and Albumen are alterred in such cases. Were these assessed and were they normal.

Minor comments

Line 47 correct spelling of deve_opes

Line 137-140 should be deleted

Author Response

Response to the comments by Reviewers:
First of all, we are thankful for your insightful review of our manuscript. We sincerely appreciate all your valuable comments and suggestions, which helped us to improve the quality of the manuscript.
Please find our responses to three reviewers.

Reviewer 1
Thank you for the opportunity to review this case report on an unusual mixture of histopathologies within an ovarian specimens. The case is clearly presented. There is logical flow of the diagnositic work up by pathology and the ultimate conclusions.
I have the following clinical questions
1.    At the Operation there was no description of the uterus and other ovary. Were these structures present or not?
2.    Was the left ovary attached to structures or freely mobile
3.    In an older woman I would have thought there would be peritoneal biopsies or node sampling/dissection? Explain the decision not to do so.
4.    Sometimes Calcium and Albumen are alterred in such cases. Were these assessed and were they normal.
Minor comments
Line 47 correct spelling of deve_opes
Line 137-140 should be deleted 

Response: Many thanks for your comments that encouraged us to further investigation/research of oncogenesis, 
Clinical questions:
1.    We have added that we observed no alterations in the uterus and other ovary at operation.
2.    The left ovary attached to structures. This has been added in the text.
3.    We did not observe peritoneal dissemination and lymph node metastases, Therefore, peritoneal biopsies or node sampling/dissection was not done.
4.    Thank you for your information. We did not find alterations in serum levels of calcium and albumin: calcium, 9.13 mg/dl (normal range: 8.8-10.1 mg/dl) and albumin, 4.25 g/dl (normal range: 4.1-5.1 g/dl). This was not added in the text.
Minor comments:
We have corrected spelling in Line 47. Sentences in Line 137-140 have been deleted.

Reviewer 2 Report

The conclusion of the paper should be more detailed. 

Author Response

First of all, we are thankful for your insightful review of our manuscript. We sincerely appreciate all your valuable comments and suggestions, which helped us to improve the quality of the manuscript.
Please find our responses to three reviewers.

Reviewer 2
The conclusion of the paper should be more detailed. 

Response: Many thanks for your suggestion, 
We have modified the conclusion in the text, as you suggested.

Reviewer 3 Report

Reviewer’s report:

The case report “Ovarian Clear Cell Carcinoma and Mature Cystic Teratoma Transformed to PNET and Carcinosarcoma: A Case Report with an Immunohistochemical Investigation” by Matsuo et al is very interesting and documents a novel finding of a probable co-existence of clear cell carcinoma and mature teratoma transformed to the somatic tumor in a 72-year-old Japanese woman. 

Here are the comments:

  • This case report is extremely rare because ARID1A-negative somatic cell-derived clear cell carcinoma coexisted with mature cyst teratoma and various tumors [primitive neuroectodermal tumor or PNET (positive for NKX2.2, CD56, and glial fibrillary acidic protein, GFAP), non-specific adenocarcinomas with AE1/AE3 positivity, and various types of sarcomas including chondrosarcoma with S100 positivity)] which were transformed from the teratoma in a single ovary with positive nuclear ARID1A.  

However, I would like to suggest adding a couple of immunohistochemical images in the main text.

  • Immunohistochemically, the authors also showed clear cell carcinoma was negative for ARID1A suggesting ARID1A mutation (which is common in endometriosis-related adenocarcinoma) and PAX8-positive staining is also suggesting its Müllerian origin.
  • Taken together, this case report is an exception and reports a complicated form of ovarian tumor derived from different mechanisms.
  • This case also highlights the significance of the appropriate immunohistochemical study to identify the cellular origin and histogenesis of ovarian cancer.

Therefore, I recommend accepting this novel case report for this esteemed journal.

Author Response

Response to the comments by Reviewers:
First of all, we are thankful for your insightful review of our manuscript. We sincerely appreciate all your valuable comments and suggestions, which helped us to improve the quality of the manuscript.
Please find our responses to three reviewers.

Reviewer 3
The case report “Ovarian Clear Cell Carcinoma and Mature Cystic Teratoma Transformed to PNET and Carcinosarcoma: A Case Report with an Immunohistochemical Investigation” by Matsuo et al is very interesting and documents a novel finding of a probable co-existence of clear cell carcinoma and mature teratoma transformed to the somatic tumor in a 72-year-old Japanese woman. 
Here are the comments:
•    This case report is extremely rare because ARID1A-negative somatic cell-derived clear cell carcinoma coexisted with mature cyst teratoma and various tumors [primitive neuroectodermal tumor or PNET (positive for NKX2.2, CD56, and glial fibrillary acidic protein, GFAP), non-specific adenocarcinomas with AE1/AE3 positivity, and various types of sarcomas including chondrosarcoma with S100 positivity)] which were transformed from the teratoma in a single ovary with positive nuclear ARID1A.  
However, I would like to suggest adding a couple of immunohistochemical images in the main text.
•    Immunohistochemically, the authors also showed clear cell carcinoma was negative for ARID1A suggesting ARID1A mutation (which is common in endometriosis-related adenocarcinoma) and PAX8-positive staining is also suggesting its Müllerian origin.
•    Taken together, this case report is an exception and reports a complicated form of ovarian tumor derived from different mechanisms.
•    This case also highlights the significance of the appropriate immunohistochemical study to identify the cellular origin and histogenesis of ovarian cancer.
Therefore, I recommend accepting this novel case report for this esteemed journal.

Response: We appreciate your warm comments. 
As you recommended we have modified Figure 4 by adding immunohistochemical images.